# Determinants of Rotavirus Vaccine Acceptance in an Area of Southern Italy with Low Vaccination Coverage: A Case-Control Study by the Health Belief Model Questionnaire

**DOI:** 10.3390/vaccines13010063

**Published:** 2025-01-13

**Authors:** Davide Anzà, Massimiliano Esposito, Giorgio Bertolazzi, Alessandra Fallucca, Carlo Genovese, Gabriele Maniscalco, Andrea D. Praticò, Tiziana Scarpaci, Ermanno Vitale, Vincenzo Restivo

**Affiliations:** 1Department of Medicine and Surgery, University of Enna Kore, 94100 Enna, Italy; davide.anza@unikore.it (D.A.); massimiliano.esposito@unikore.it (M.E.); giorgio.bertolazzi@unikore.it (G.B.); carlo.genovese@unikore.it (C.G.); andrea.pratico@unikore.it (A.D.P.); ermanno.vitale@unikore.it (E.V.); 2Department of Health Promotion, Maternal and Infant Care, Internal Medicine and Medical Specialties (PROMISE) “G. D’Alessandro”, University of Palermo, 90127 Palermo, Italy; alessandra.fallucca@unipa.it (A.F.); gabriele.maniscalco02@community.unipa.it (G.M.); tiziana.scarpaci@community.unipa.it (T.S.)

**Keywords:** rotavirus, vaccination coverage, health belief model, acute gastroenteritis, hospital admission, parents acceptance, vaccine refusal

## Abstract

Background/Objectives: Rotavirus (RV) is the primary cause of gastroenteritis in children worldwide, contributing significantly to morbidity and mortality, particularly among children under five years of age. The introduction of Rotavirus vaccines (RVV) has markedly reduced RV-related childhood deaths, especially in Europe, where substantial reductions in hospitalizations and disease prevalence have been observed. Despite these advances, RVV uptake in Italy remains below the desired targets, with notable regional disparities. In Sicily, vaccination rates have fluctuated, with current coverage failing to meet national goals. Safety concerns and insufficient parental awareness are major barriers to RVV acceptance. Methods: This case-control study was conducted in Southern Italy to identify factors influencing parental acceptance of RVV. Data were collected from parents using a structured questionnaire that assessed socio-demographic factors, vaccine knowledge, and attitudes based on the Health Belief Model (HBM). Results: Overall, 226 parents were enrolled. Higher perceived benefit of RVV was significantly associated with increased vaccine adherence (Odds Ratio = 13.65; 95% Confidence Interval = 6.88–27.09; *p* < 0.001). Conclusions: These results highlight the need for targeted interventions to improve vaccine coverage and address regional and socio-economic barriers to RVV acceptance. Furthermore, tailored educational campaigns and univocal information from healthcare providers could play pivotal roles in achieving higher vaccine uptake.

## 1. Introduction

Rotavirus (RV) continues to pose a significant global health challenge as the leading cause of severe gastroenteritis worldwide, particularly among children under five years of age. Each year, RV is responsible for nearly 258 million infections and 129,000 deaths in this vulnerable group [1]. However, since the approval of Rotavirus vaccines (RVVs) in 2006, substantial progress has been achieved, with the number of childhood deaths from RV infections significantly decreasing worldwide, dropping from over 500,000 per year in 2000 to 125,000 in 2018 [2]. In Europe, before the introduction of RVV, the burden of Rotavirus gastroenteritis (RVGE) was considerable, accounting for approximately 3.6 million cases annually in children under five years, leading to 87,000 hospitalizations and 700,000 pediatric visits [3]. In Italy, RV circulates throughout the year, with a higher incidence from October to May and peaks occurring one to two months later in the southern regions compared to the northern and central regions [4]. In addition, RVGE cases account for more than a third of all hospitalizations for acute gastroenteritis (AGE), particularly in infants aged 3 to 36 months [5,6].

To prevent infection and reduce disease severity, the World Health Organization (WHO) recommends the inclusion of live attenuated RVV in all national vaccination programs [7]. These vaccines have been authorized for use after being found safe and effective in clinical trials [8,9]. Furthermore, several pieces of research on RVVs have demonstrated up to 40% reduction in disease prevalence in countries participating in the global RV surveillance network [10], as well as 86% effectiveness among children younger than 12 months [11], proving a high level of protection.

In Italy, universal vaccination against RV has been included in the National Vaccination Prevention Plan (PNPV) since 2017, making it available and free of charge for all children 6 weeks of age or older [12]. Two live attenuated vaccines are available in Italy: the monovalent administrable in two doses and the pentavalent recombinant human-bovine administrable in three doses. These vaccines can be administered starting from 6 weeks of age to 24 or 32 weeks of age, respectively [13].

According to the Italian Ministry of Health, the current vaccination coverage against RV in children up to 24 months is 74% [14], although there are significant regional differences. In addition, the target of reaching a vaccination coverage rate of at least 95% by 2020 has still not been met. Among the Italian administrative regions, Sicily with 35,000 newborns per year, became the first to include RVVs in its newborn vaccination program in January 2013 [15]. This initiative led to an immunization rate of approximately 45%, resulting in a 47% reduction in hospital admissions due to RVGE [16]. In detail, the RVGE hospitalization rate decreased from 394 per 100,000 in 2009–2012 to 220 per 100,000 in 2013–2016 [16]. However, despite this pioneering step towards universal free immunization, after five years of active and free administration of the vaccine (2018), the average vaccination coverage against RVGE in Sicily has fallen to below 40%, albeit with significant differences in coverage rates between western (over 48%) and the eastern administrative provinces (under 34%) [17]. Currently, RVV coverage in Sicily is 64.4% [13], failing to meet the target set by the Italian Ministry of Health [12]. Additionally, there is still a general lack of information regarding the reasons for the low adherence to RVV [18,19]. Historically, concerns regarding the safety of RVVs have been raised, particularly with the first-generation vaccine released in 1999, which was associated with an increased risk of intussusception [20,21,22]. However, post-licensure surveillance data from Europe for the new generation RVVs strongly supported their favorable safety profiles, demonstrating that the risk of intussusception was minimal when the vaccine was administered early, within the recommended age range of 6–8 weeks [23,24]. Notably, in Sicily, a study investigating the risk factors for intussusception (RAIS) found no association between RVVs and intussusception in children aged 0–59 months [25], which was in line with the results of similar studies conducted in other countries [26]. Moreover, the World Health Organization (WHO) has highlighted that the potential risk of intussusception is exceptionally low, estimating an increase of only 1–2 additional cases per 100,000 vaccinated infants [27]. The WHO further emphasizes that the significant benefits of vaccination, including a marked reduction in severe RV infections and related hospital admissions, vastly outweigh the minimal risks associated with RVVs [27].

According to the literature, the low RVV coverage should be related to both healthcare worker recommendations and factors related to eligible parents [28,29]. While healthcare workers’ roles in influencing vaccine acceptance are well documented, limited investigations have been conducted on reasons affecting parental decisions, as well as their low RVV acceptance.

The main objective of this study was to identify key factors influencing parental compliance with RVV in an area of Italy with low vaccination coverage.

## 2. Materials and Methods

### 2.1. Study Design

This case-control study was conducted to explore factors associated with RVV acceptance. Cases were defined as children fully vaccinated against RV between 2 and 8 months of age, while controls included children of the same age who had not been immunized with RVV. The data were collected between December 2022 and September 2023 in the administrative province of Palermo (Sicily, Italy). The target population was identified according to the RVV status through the information system of the Local Health Unit of Palermo. This is the electronic record that collects data of all people registered in the civil registry.

### 2.2. Sample Size

The sample size was evaluated using the formula n = 1.96^2^ × proportion × (1 − proportion)/accuracy. The calculation was based on the expected positive attitude of parents towards the willingness to vaccinate their children and reaching the objective of the Italian PNPV coverage, with an RVV coverage of 85% [13,19,28], a confidence interval (CI) of 95%, and an accuracy of 5%. According to this formula, the sample size was 196 individuals. Assuming a refusal of 10%, the final sample size was calculated as 216 individuals.

### 2.3. Questionnaire

A questionnaire was designed and administered via telephone to parents who had been fully informed about the survey’s purpose and the measures taken to ensure data confidentiality. The questionnaire was divided into three sections. The first collected were socio-demographic and health data, including the child’s date of birth, gender, health status, parent’s education level, employment status, parents’ age, household composition, and the number of other children. The second section of the questionnaire addressed parents’ knowledge of the vaccine, willingness to recommend vaccination to others, sources of information, and breastfeeding practices. The third section explored the acceptance of RVV based on the Health Belief Model (HBM). The four domains explored with the HBM were the perceived susceptibility, the severity of RV infections, the perceived barriers, and the benefits of RVV. Parents indicated their agreement by answering five-point Likert scale questions (Table 1), ranging from “strongly agree” to “strongly disagree”. The responses were converted to an ordinal scale, with 1 indicating strong disagreement and 5 indicating strong agreement. Before the beginning of the study, the questionnaire was validated in a convenience sample representing approximately 10% of the parents.

### 2.4. Statistical Analysis

A P-value of 0.05 was considered statistically significant (two-tailed) for all analyses. The normality of the quantitative variable distributions was evaluated using the Skewness and Kurtosis tests. Quantitative variables with a normal distribution were reported as mean (standard deviation), while those with a non-normal distribution were reported as median (interquartile range). For qualitative variables, both absolute and relative frequencies were calculated to provide a comprehensive overview of the data. The relationships between adherence to RVV and normally or non-normally distributed quantitative variables were analyzed using the Student’s T-test or the Wilcoxon and Mann–Whitney tests, respectively, depending on the distribution. Furthermore, relationships with qualitative variables were assessed using the Chi-square test (χ^2^). Finally, all variables significantly associated with adherence to RVV were included in a multivariate logistic regression model to adjust for confounding factors. Sex and age (in months) of children were included in the multivariate model as a priori confounders. All analyses were performed using the statistical software Stata MP 14.2 (StataCorp LLC, 4905 Lakeway Drive, College Station, TX 77845, USA), ensuring the reliability and precision of the results.

## 3. Results

### 3.1. Descriptive Characteristics

A total of 290 eligible parents were contacted and 226 agreed to participate in this study (participation rate 78%). Among the participants’ parents, 49% (n = 110) had RVV from their children. Table 2 describes the socio-demographic data of the children. In particular, the newborns were between 2 and 5 months old (median of age 4 months), about half (53.5%, n = 121) were male, while 65.8% (n = 110) were vaccinated against RV. Among the parents, those aged between 25 and 35 years were prevalent (55%, n = 125). About half of the respondents lived in the urban area of Palermo, the majority were housewives (37.1%), and about 40% lived with two other people.

The χ^2^ revealed a significant difference in the variable “cohabitation” for respondents in the unvaccinated cohort with two cohabitants (42.7% vs. 38.8%, *p* = 0.03) and for those in the vaccinated cohort with three cohabitants (43.1% vs. 35.5%, *p* = 0.03).

### 3.2. Knowledge of Anti-Rotavirus Vaccine

The results of the second section of the questionnaire (Table 3) indicate that 79.3% (n = 179) of respondents were aware of the RVV for infants. More parents (69.4%, n = 157) declared that the primary source of health information was healthcare professionals (68.6%, n = 155), the preferred source of information about vaccination was their pediatrician (69.4%, n = 157), while 74% (n = 167) of them would recommend the RVV to others. Regarding breastfeeding practice, only 31.4% (n = 71) were exclusively breastfed, while 55.3% (n = 125) of newborns were artificially fed formula milk. Parents who refused the RVV had a higher rate of knowledge related to vaccine availability than the RVV acceptors (86.4% vs. 72.4%, *p* = 0.01). On the other hand, parents who accepted RVV would recommend the vaccine more frequently than parents who did not (97.4% vs. 49.1, *p* < 0.001).

### 3.3. Acceptance of Anti-Rotavirus Vaccine by Health Belief Model

The results regarding the HBM model showed that perceived severity had a higher median score (median = 9), followed by perceived barriers and perceived benefits (median = 8). The domain with the lowest median score was perceived susceptibility (median = 7). No differences were found between the two cohorts in perceived susceptibility (*p* = 0.51) and perceived severity (*p* = 0.85). On the other hand, a significant difference between the two cohorts was found for the variables “high perceived barriers” (38.8% vs. 25.4%, *p* = 0.03) and “high perceived benefits” (75.9% vs. 18.2%, *p* < 0.001) (Table 4).

In the multivariate analysis (Table 5), the only factor significantly associated with RVV adherence was a higher perception of vaccine benefits by parents (adjusted Odds Ratio [OR] = 13.65; 95% CI = 6.88–27.09; *p* < 0.001).

## 4. Discussion

As a consequence of the introduction of RVV into childhood immunization programs, several European countries have reported significant public health benefits, such as a reduction in the incidence of RVGE [2], as well as a substantial decline in healthcare utilization, impacting on decreased hospitalizations, nosocomial RV infections, and outpatient visits related to RV infections [16]. Such outcomes highlight the transformative impact of RVVs in mitigating the burden of RV-related diseases and reducing associated healthcare costs. While progress has been made, there are still notable disparities in vaccine uptake in Italy. Sicily, for example, was the first Italian administrative region to implement a universal mass vaccination program against RV in 2013 [15], demonstrating early commitment to reducing the public health impact of RV. However, despite this pioneering step, the immunization rate remains only 64.4% as of the latest data [17], well below the national target of achieving a vaccination coverage rate of ≥95% [12].

Limited information is available regarding the factors influencing parental acceptance or refusal of RVV. This study identified the perceived benefit of RVV as the only significant factor associated with higher vaccination adherence (adjusted OR = 13.65; 95% CI = 6.88–27.09). To date, only a single cross-sectional study, conducted in 2022, has investigated the reasons behind low RVV adherence among parents. This earlier study reported a 15.3% adherence rate, with fewer than 50% of parents being aware of RV infection and only 59% demonstrating knowledge about RVV [30]. Moreover, the survey underscored a pronounced parental reluctance to vaccinate their children against RV, despite widespread concerns about the potential severity of RVGE [30]. However, this prior research did not incorporate a cognitive model to systematically explore the factors influencing parental decision-making.

On the other hand, the current study used the HBM model, according to which the perceived benefit was built up with two questions: perception of efficacy and safety of RVV. Parental confidence in the efficacy of RVV may be related to other factors such as education level, vaccine knowledge, and thorough consultation by the pediatrician. Nevertheless, the effectiveness of modern RVVs has been confirmed by clinical trials and surveys, which documented an efficacy of 86% among children younger than 12 months and sustained high level of protection [11,31]. Furthermore, the RVV seems to have high effectiveness even if vaccination coverage was under the recommended level. For example, a study conducted in Southern Italy showed that 35% of vaccination coverage caused a reduction in hospital admission of 39% in comparison to the period without vaccination availability [16].

Further, regarding safety, all vaccine adverse events are consistently tracked in the continuous market surveillance studies. Indeed, previous surveillance studies of RVV have found an increased risk of intussusception in some infants and young children associated with the first-generation oral tetravalent RVV, which was subsequently withdrawn from the market in 1999 [32]. New RVVs are highly safe since numerous studies have shown no association between the incidence of intussusception and RVV [25,33]. Moreover, it is crucial to emphasize that the benefits of vaccination significantly outweigh the vaccine’s potential risks [25].

Awareness of RVV safety and appropriate counseling may strongly correlate with lower concern about adverse reactions and thus improve vaccination adherence [34]. Furthermore, it is well established that pediatricians’ advice and recommendations provide crucial information on the administration of vaccines, including the optimal age for vaccination and detailed data on the disease caused by the virus and its manifestations among infants. Indeed, pediatricians are educated about the safety and effectiveness of preventive strategies such as vaccination during their specialty school education, in Italy. In our study, pediatricians were explicitly declared as the most frequently mentioned source of information on RVV (69%), demonstrating that pediatricians play a crucial role in protecting individual children’s health and involving parents in making informed decisions. From a public health perspective, pediatricians should be central in increasing vaccination coverage with comprehensive, clear, and understandable information about RV infection, the safety of the vaccine, and possible side effects.

Online platforms also play a growing role in shaping public opinion. Therefore, around a third (26%) of participants stated that they use websites and social media to obtain health-related information for themselves and their children. While these sources can be quick, they should be approached cautiously as they can spread misinformation and potentially fuel distrust of vaccines. Thus, monitoring social media for anti-vaccination sentiment is crucial. The web listening technique, for example, could be a plausible way of capturing the nature and direction of online discourses about vaccination and could be helpful in understanding how social media can influence public perceptions and therefore decisions about vaccination [35]. Furthermore, online platform monitoring may play a fundamental role in shaping public vaccination perceptions and decisions by effectively addressing user concerns and promoting informed decision-making helped by healthcare workers [35]. Effective health communication strategies must not only counter anti-vaccination campaigns but also provide users with transparent, evidence-based information that is easily accessible. Another proposal to increase the effective use of the web should be to inform people to use health-certified websites. For example, the Health on the Net Foundation code certification ensures information reliability and compliance with ethical standards, promoting user trust. Conversely, uncertified sites may not report the publication date, the date of the last update, or the scientific quality of the source, which can lead to uncertainty about the timeliness and relevance of the information provided [36].

Current strategies to address vaccine hesitancy are limited. However, according to the WHO Strategic Advisory Group of Experts on Immunization working group, multi-component, dialogue-based interventions are considered the most effective [37,38]. These strategies include training healthcare workers to manage parental vaccine hesitancy effectively and address it within their ranks [37]. To overcome vaccination hesitancy in the context of childhood, it is also crucial to underscore the importance of understanding parents’ perspectives to enhance communication between healthcare professionals and parents. Specifically, a study conducted in Rome (Italy) offered valuable insights for developing interventions to reduce vaccination hesitancy among pregnant women. The study highlighted pregnancy as a strategic “teachable moment” for health promotion and behavior change, with the prenatal period being a critical time when attitudes toward vaccination are generally explored and solidified [39]. Therefore, pregnant women represent an ideal population for targeted interventions to increase vaccine awareness and confidence [37,40]. Other Italian studies found that mothers are even 1.5 times more likely than fathers to be the primary decision-makers regarding their children’s immunizations [41,42]. This pattern is particularly pronounced in Italy, where women are predominantly responsible for family care and health [41]. Regarding our study, a larger proportion of parents interviewed were housewives (37%), which is interesting as they may represent the optimal target group for information campaigns aimed at increasing childhood immunization coverage, as well as be a more reliable source of information when it comes to assessing children’s immunization status. To improve the knowledge of pregnant women about RVV a coordinated and unambiguous information campaign by all healthcare workers involved in the pregnancy pathway (as gynecologists, obstetricians, nurses, pediatricians, general practitioners, pharmacists, etc.) should be conducted.

The present study has several limitations that should be considered. First, the recall bias could reduce the quality of information. Secondly, the study sample, limited to the administrative province of Palermo, may not be representative of the broader population, thereby potentially limiting the generalizability of the findings. However, this is one of the first studies to examine RVV acceptance from parents’ perspective using a cognitive model, providing valuable insights into the factors influencing vaccine adherence and monitoring during time the phenomenon. Additionally, the vaccination status of the infants, obtained through direct interviews with the parents, was corroborated using the vaccination registry. This rigorous approach ensured the accuracy of the vaccination data, effectively minimizing the potential for errors and enhancing the reliability of the results.

## 5. Conclusions

To significantly implement the elimination of RV, it is helpful to understand why parents choose not to vaccinate their children. This study’s findings emphasize that the perceived benefits of vaccination, such as awareness of the safety and efficacy of RVV, are key factors influencing parents’ compliance. Therefore, the counseling about RVV led by healthcare professionals should be tailored to parents’ needs, focusing on the benefits of RVV.

## Figures and Tables

**Table 1 vaccines-13-00063-t001:** Questions of the Health Beliefs Model about Rotavirus vaccination acceptance.

Perceived susceptibility	(a) Children are more likely to catch RVGE in the first few days of life. (b) It is easy to contract RVGE by living with other people and/or attending nurseries.
Perceived severity	(c) RVGE can lead to complications (such as diarrhoea, and shock) and/or hospitalization. (d) Children with RVGE require parents’ support, and it can cause loss of workdays.
Perceived barriers	(e) I believe that I can easily access vaccination services to receive the RVV. (f) RV vaccination can be administered even in children with neonatal colic.
Perceived benefits	(g) RVV is effective in preventing RVGE. (h) RVV has no severe adverse reactions.

**Table 2 vaccines-13-00063-t002:** Socio-demographic information of the children.

	Total (n = 226)	Parents Who Refused RVV (n = 116, 51.3%)	Parents Who Accepted RVV (n = 110, 48.6%)	*p*-Value
Gender				
Males	121 (53.5%)	58 (52.7%)	63 (54.3%)	0.81
Females	105 (46.4%)	52 (47.3%)	53 (45.7%)	
Median age in moths	4 (3–4)	4 (3–5)	4 (3–4)	0.07
Pathology of children				
Yes	9 (4.0%)	7 (6.4%)	2 (1.7%)	0.07
No	217 (96.0%)	103 (93.6%)	114 (98.3%)	
Residence				
Province of Palermo	113 (50%)	54 (49.1%)	59 (50.9%)	0.79
City of Palermo	113 (50%)	56 (50.9%)	57 (49.1%)	
Parents’ Occupation				
Employee	75 (33.1%)	32 (29.1%)	43 (37.1%)	0.08
Self-employed	13 (5.7%)	4 (3.6%)	9 (7.7%)	
Healthcare worker	11 (4.8%)	8 (7.3%)	3 (2.6%)	
Unemployed	39 (17.2%)	25 (22.7%)	14 (12.1%)	
Housewife	84 (37.1%)	40 (36.4%)	44 (37.9%)	
Other	4 (1.7%)	1 (0.9%)	3 (2.6%)	
Family Cohabitants				
1	6 (2.7%)	5 (5.6%)	1 (0.9%)	0.03
2	92 (40.7%)	47 (42.7%)	45 (38.8%)	
3	89 (39.4%)	39 (35.5%)	50 (43.1%)	
4	24 (10.6%)	13 (11.8%)	11 (9.5%)	
>4	15 (6.6%)	6 (5.5%)	9 (7.8%)	
No. of sons	2 (1–2)	1 (1–2)	2 (1–2)	0.98

**Table 3 vaccines-13-00063-t003:** Parents’ knowledge about the Rotavirus vaccine.

	Total (n = 226)	Parents Who Refused RVV (n = 116, 51.3%)	Parents Who Accepted RVV (n = 110, 48.7%)	*p*-Value
Knowledge of rotavirus vaccine				
No	47 (20.7%)	15 (13.6%)	32 (27.6%)	0.01
Yes	179 (79.3%)	95 (86.4%)	84 (72.4%)	
The main source of information about the rotavirus vaccine				
General practitioner	7 (3%)	4 (3.6%)	3 (2.6%)	0.47
Pediatricians	157 (69.4%)	71 (64.6%)	86 (74.1%)	
Web	4 (1.7%)	2 (1.8%)	2 (1.7%)	
Others sources	58 (25.6%)	33 (30%)	25 (21.6%)	
The main source of health information				
Scientific books/papers	1 (0.4%)	1 (0.9%)	0 (0%)	0.5
Friends or relatives	8 (3.5%)	2 (1.8%)	6 (5.2%)	
Web	56 (24.7%)	30 (27.3%)	26 (22.4%)	
Social networks	3 (1.3%)	2 (1.8%)	1 (0.9%)	
Healthcare workers	155 (68.6%)	73 (66.4%)	82 (70.7%)	
Others	3 (1.3%)	2 (1.8%)	1 (0.9%)	
Neonatal feeding				
Breastfeeding	71 (31.4%)	30 (27.3%)	41 (35.3%)	0.35
Mixed	30 (13.2%)	14 (12.7%)	16 (13.8%)	
Artificial	125 (55.3%)	66 (60%)	59 (50.9%)	
Recommended vaccination to other parents				
No	59 (26.1%)	56 (50.9%)	3 (2.6%)	<0.001
Yes	167 (73.8%)	54 (49.1%)	113 (97.4%)	

**Table 4 vaccines-13-00063-t004:** Health Belief Model of the parents.

	Total (n = 226)	Parents Who Refused RVV (n = 116, 51.3%)	Parents Who Accepted RVV (n = 110, 48.7%)	*p*-Value
Perceived susceptibility				
Mean score (SD)	6.6 (1.9)	6.7 (1.8)	6.6 (1.9)	0.57
Low (<median)	118 (52.2%)	55 (50%)	63 (54.3%)	0.51
High (>median)	108 (47.7%)	55 (50%)	53 (45.7%)	
Perceived severity				
Mean score (SD)	8.0 (1.8)	8.0 (1.8)	8.1 (1.7)	0.65
Low (<median)	128 (56.6%)	63 (57.3%)	65 (56%)	0.85
High (>median)	98 (43.3%)	47 (42.7%)	51 (44%)	
Perceived barriers				
Mean score (SD)	6.7 (1.9)	6.4 (1.9)	6.9 (1.9)	0.05
Low (<median)	153 (67.6%)	82 (75.6%)	71 (61.2%)	0.03
High (>median)	73 (32.3%)	28 (25.4%)	45 (38.8%)	
Perceived benefits				
Mean score (SD)	7.0 (2.2)	5.7 (2.1)	8.3 (1.5)	<0.001
Low (<median)	118 (52.2%)	90 (81.8%)	28 (24.1%)	<0.001
High (>median)	108 (47.7%)	20 (18.2)	88 (75.9%)	

**Table 5 vaccines-13-00063-t005:** Univariable and multivariable analysis of factors associated with anti-Rotavirus vaccine acceptance.

	Univariable Analysis			Multivariable Analysis		
	Naive OR	CI 95%	*p*-Value	Adjusted OR	CI 95%	*p*-Value
Male vs. female	0.93	0.55–1.58	0.81	0.79	0.39–1.56	0.49
Age per month increase	0.77	0.59–1.01	0.06	0.99	0.72–1.37	0.96
Urban vs. province	0.93	0.55–1.56	0.79	-	-	-
Pathology vs. no pathology	0.25	0.05–1.27	0.09	-	-	-
Self-employed vs. employee	1.67	0.47–5.92	0.42	1.36	0.29–6.45	0.69
Healthcare worker vs. employee	0.27	0.06–1.13	0.07	0.26	0.49–1.41	0.12
No occupation vs. employee	0.41	0.18–0.92	0.03	0.54	0.2–1.46	0.22
Housewife vs. employee	0.81	0.43–1.53	0.53	1.2	0.54–2.67	0.65
Other work vs. employee	2.23	0.22–22.47	0.49	1.40	0.10–18.56	0.8
2 vs. 1 cohabitant	4.78	0.53–42.58	0.16	-	-	-
3 vs. 1 cohabitant	6.41	0.71–57.1	0.09	-	-	-
4 vs. 1 cohabitant	4.23	0.42–41.8	0.21	-	-	-
>4 vs. 1 cohabitant	7.49	0.69–81.2	0.09	-	-	-
The number of sons per son increases	1.2	0.86–1.66	0.26	-	-	-
Vaccine knowledge: yes vs. no	0.41	0.2–0.81	0.01	-	-	-
Vaccine information source: Pediatrician vs. General Practitioner	1.61	0.34–7.45	0.53	-	-	-
Web vs. General Practitioner	1.33	0.11–15.7	0.81	-	-	-
Other sources of information vs. General Practitioner	1.01	0.21–4.93	0.99			
Health information source				-	-	-
Parents or relatives vs. other sources	5.99	0.33–107.4	0.22	-	-	-
Web vs. other sources	1.73	0.14–20.2	0.66	-	-	-
Social networks vs. other sources	1	0.03–29.8	1	-	-	-
Healthcare workers vs. other sources	2.24	0.19–25.2	0.51	-	-	-
Recommendation of the vaccine: Yes vs. no	39.06	11.6–130.4	<0.0001	-	-	-
Breastfeeding: Mixed vs. Breastfeeding	0.83	0.35–1.97	0.68	-	-	-
Artificial vs. Breastfeeding	0.65	0.36–1.17	0.15	-	-	-
Higher vs. lower perceived susceptibility	0.84	0.49–1.41	0.51	0.96	0.48–1.95	0.92
Higher vs. lower perceived severity	1.05	0.62–1.78	0.85	1.09	0.53–2.23	0.82
Higher vs. lower perceived barriers	1.85	1.05–3.27	0.03	1.53	0.72–3.25	0.27
Higher vs. lower perceived benefits	14.1	7.42–26.9	<0.001	13.65	6.88–27.09	<0.001

## Data Availability

Data will be available upon a motivated request to the corresponding author.

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
