# Peer review of "Determinants of Rotavirus Vaccine Acceptance in an Area of Southern Italy with Low Vaccination Coverage: A Case-Control Study by the Health Belief Model Questionnaire"

_vaccines, 2025, doi:10.3390/vaccines13010063_

Round 1
Reviewer 1 Report
Comments and Suggestions for Authors
I would like to congratulate the authors for their study using the HBM model to explain why RVV uptake in Sicily, Italy remains below the desired targets.
They conducted a case-control study presenting a questionnaire to evaluate the social psychological health behavior change of parents in regard to the uptake of the RVV.
The authors explained, determined and suggest strategies to address vaccine uptake taking into account different parameters.
NB. Please corrections should be made on Table 2 where the percentage of children who refused the rotavirus vaccine and those who accepted should be corrected to 51.32% and 48.67% respectively.
Author Response
I would like to congratulate the authors for their study using the HBM model to explain why RVV uptake in Sicily, Italy remains below the desired targets.
They conducted a case-control study presenting a questionnaire to evaluate the social psychological health behavior change of parents in regard to the uptake of the RVV.
The authors explained, determined and suggest strategies to address vaccine uptake taking into account different parameters.
Please corrections should be made on Table 2 where the percentage of children who refused the rotavirus vaccine and those who accepted should be corrected to 51.32% and 48.67% respectively.
Answer:
Thank you for showing the mistake made in the percentage and we revised Table 2 accordingly. This change can be found in the revised manuscript on Table 2-3-4 at pages 4,5,6.
Reviewer 2 Report
Comments and Suggestions for Authors
This study represents solid and important work analyzing vaccine acceptance/hesitancy in Sicily. Although the study’s results are not surprising, they emphasize the need for such work and how, while general patterns for this issue might be consistent, each region inevitably has its unique set of cultural values and knowledge that affect vaccine use. Thus, the more work like this the better. The authors do a good job of shaping a survey that efficiently captures the key issues and questions, and their results and proposed solutions are well contextualized within the national and international literature. My only minor concerns are related to the English, especially in Table 1 and throughout the Discussion, but these are at a level that can be easily clarified by language editors.
Comments on the Quality of English Languagesee above
Author Response
This study represents solid and important work analyzing vaccine acceptance/hesitancy in Sicily. Although the study’s results are not surprising, they emphasize the need for such work and how, while general patterns for this issue might be consistent, each region inevitably has its unique set of cultural values and knowledge that affect vaccine use. Thus, the more work like this the better. The authors do a good job of shaping a survey that efficiently captures the key issues and questions, and their results and proposed solutions are well contextualized within the national and international literature. My only minor concerns are related to the English, especially in Table 1 and throughout the Discussion, but these are at a level that can be easily clarified by language editors.
Answer:
Thank you for the comments and we revised table 1 and discussion section according to your suggestions.
Reviewer 3 Report
Comments and Suggestions for Authors
Please review my comments in the following upload- thank you for your efforts to help or children

Author Response
Introduction-project rationale/ Aims: The reported study is a public health project conducted by a group of Italian academicians who are based in two Italian Universities. The authors provide and in-depth description of the current state of knowledge and benefits and unmet needs of the use of the Rota virus vaccine. They have discovered that vaccine uptake within Italy and in certain regions is not optimal leading to their specific project aims. As stated in their manuscript- “However, despite this pioneering step towards universal free immunization, after five years of active and free administration of the vaccine (2018), the average vaccination coverage against RVGE in Sicily has fallen to below 40%, albeit with significant differences in coverage rates between western (over 48%) and the eastern administrative provinces (under 34%) [16]. Currently, RVV coverage in Sicily is 64.4% [17], failing to meet the target set by the Italian Ministry of Health [12]. Additionally, there is still a general lack of information regarding the reasons for the low adherence to RVV [18,19].” This lays the foundation for their project in which they used the well-recognized Health Benefits Belief Model using a questionnaire as their primary data collection instrument. The following are specific section comments:
- Material and methods- the authors have focused their efforts on the region of Palermo in Sicily. They are in that region and Palermo was on the vanguard introducing the vaccine. They state the vaccine uptake is far below the goals and the vaccination rate appears to be dropping.
- Sampling and samples size- the authors determined a rather small sample size and do not provide a global number of children within this region that are potential subjects for the study. Their final number is only ~200 but provide sample size justification based on power calculations related to a positive parenteral response. It would be informative to know the number of reported total area potential vaccine candidates, total RVV rate within the region and refusal. Also report any regional differences granted it is in one region. Additional specific should be provided on how the subjects were identified contacted and consented as only a generic statement was provided – “through the information system of the Local Health Unit of Palermo” Does this system cover all Palermo? Is this an Electronic Health record?
Answer: Thank you for your revision which could increase the quality of the manuscript. The total population that potentially could be vaccinated is newborns. Every year there are around 35,000 newborns in Sicily. When the study started Rotavirus vaccination coverage was 64% in Sicily with missed vaccination of 36%. However, no data about how many missed vaccination was related to refusal or other reasons are not monitored by the Local Health Unit. The identification of subjects eligible for the study started from the electronic record of the Palermo Health Unit which collects data of all people registered in the civil registry. The system covers all Palermo administrative people living in the province and it is an electronic record that all healthcare workers the vaccination services can implement. Subsequently, parents who have children with at least one vaccination recorded but not the Rotavirus vaccine were considered as potential cases. On the other hand, parents who have a child vaccinated for Rotavirus with the same characteristics of potential cases were considered potential control. We added details in the introduction and material and method section to explain it.
- Human research studies protection – The authors state the following: “A questionnaire was developed and then administered by telephone to the parents who were adequately informed about the purpose of the survey and the methods used to 89 ensure the confidentiality of the data. This statement raises some concerns at least from the context of IRB issues. How were the subjects recruited by contact? Did subjects consent to being called? “Cold calling” in not approved in studies in the US and the authors should provide 1 explanation in detail as to their recruitment specifics. Cold calling is considered intrusive, disruptive and may cause stress in the subjects. When the subject was called how specifically was the study described and how was consent recorded and validated? Contact by phone and consenting subjects should be guided by an IRB reviewed and approved script. This is to assure non-coercion and to provide consistent recruitment efforts. The authors should provide any script in an appendix translated into English. Authors should provide a copy of the consent documentation record.
Answer: As reported in the Patents section of the manuscript the study was already approved by the Ethical Committee of Palermo 1 as follows “Institutional Review Board Statement: The study was conducted in accordance with the Declaration of Helsinki and approved by the Institutional Review Board (or Ethics Committee) of Palermo 1 (protocol code 08/2021 approved on 15 September 2021).” Furthermore, parents were informed that they could be potentially contacted for preventive or research reasons because they had an informative and consent for each vaccination that explained this possibility. According to your request, we add in the appendix the translated informative statement and consent of the study. Moreover, we add to you the consent section used by the vaccination services where is reported the possibility of being contacted for research reason.
- Questionnaire- The authors state they developed a local developed questionnaire. It appears to be innovative. The concern is there is no evidence this instrument had any validation steps taken or determine content, internal or external validity conducted. As such it is unknown if this questionnaire is a valid instrument for subject data collection. If established steps were taken for validation they should be described in the manuscript. If the questionnaire was used from previous studies, please describe as I could not find a linking reference.
Answer: The questionnaire used in the research has been submitted to an external validation process to understand the comprehension of the questions. The following sentence was added to the method section “Before the beginning of the study, the questionnaire was validated in a convenience sample representing approximately 10% of the parents.” Furthermore, the section related to the belief about Rotavirus vaccination was constructed in accordance to the HAPA model as already explained in the method section.
- Results-
- Table 2-Small point but in the description titles- children don’t refuse the vaccine it’s their parents- so just relabel vaccine acceptors versus vaccine acceptors?
Answer: The headings of the tables 2, 3 and 4 were modified according to your suggestion.
- Are respondent housewives “unemployed” as should they just be classified within that category? In that context does unemployed mean no one in the family has a supportive job? Please clarify
Answer: Housewives are not classified as unemployed for two reasons. Firstly, women classified as unemployed are searching for work differently from housewives who are not searching for work because they already had a work and consequently can have access to a different source of information for health topics. Secondly, housewives can have a retirement benefit in Italy and consequently they have a different socio-economic level of women who are unemployed. For these reasons we decided to maintain a difference for these two types of occupation.
- Perhaps replace “freelancer” with “self-employed?”
Answer: Thank you for your suggestion we replaced the word.
Page 5- The following verbiage is confusing as to the meaning- please rewrite to clarify: Exploring differences by anti-RVV status, parents who refused the vaccine declared more to know the availability of vaccination (86.4% vs 72.4%, p=0.01). Suggest- Parents that refused the RVV had a higher rate of knowledge related to vaccine availabily than the RVV acceptors. On the other hand, parents who accepted anti-RVV would recommend the vaccine more frequently than parents who did not 140 (97.4% vs. 49.1, p<0.001). The term anti-RVV is confusing and counterintuitive. Just state RVV- Its understood this is referring to the Rota Virus Vaccine.
Answer: We followed your suggestions to rephrase the sentence and adopt a unique term for RVV.
Table 4 uses numbers and % above and below the median of the scoring. Its somewhat unconventional and difficult to determine if this categorical difference (below versus above median) has validity as a statistical measure. I would suggest this be replaced with the actual mean SD data for stronger direct discrimination between groups.
Answer: We added in table 4 the mean score for each domain to increase the comprehension of the score difference.
Table 5- See the pasted info. These two variables are not defined in the manuscript
Age per month increase
Pathology vs no pathology
Answer. We defined the variable pathology related to health status of children in the material and method section and added it in table 1. On the other hand, the age per month of children was already defined in the material and method section.
The only positive finding is that RVV acceptors were those that have a high acceptance of benefits. This is a finding that is intuitive bur reinforces validity of the survey. I suspect the small sample size, and the many comparisons dilutes the strength of the other analyses.
Answer: We stated in the limits section that the small sample size could reduce the generalizability of the data but we think that the analysis showed the most important associated factor with vaccine acceptance and other studies can have a different result because the setting or people belief living in other area can influence the association.
- Discussion- is an in-depth analysis of their results benchmarked against existing studies. Some comments a. Source of health information- as described and appropriate that the primary health information is from Pediatricians. The question is how consistent Italian Pediatricians in are providing accurate and positive recommendations for vaccinations. This is often a “missing link” in vaccine studies. How does Italy educate health care providers in educating their patients on the significant benefits and extreme low risk of the RVV? The accessing of health information from the WEB/ social media is important public health finding but unfortunately the project did not get what WEB/social media sites were accessed. This is a critical issue based on current global problem with health misinformation. The authors do provide reasonable solutions addressing this problem in general.
Answer: We increased the discussion section about pediatricians as source of information including a sentence where it is described their education about vaccination topic “Indeed, pediatricians are educated about the safety and effectiveness of preventive strategies, such as vaccination, during their specialty school education, in Italy”. For the web or social media source of information, we highlighted two proposals addressing the problem: web monitoring and information campaign about the safe use of the web which are detailed more in the discussion section.
- Parent education- the authors site important information on educating pregnant women on the need for vaccination and this reviewer agrees this may be one of the critical times to discuss risk and benefits. No direct information however was determined with the subjects related to pregnancy education related to the RVV. I believe this relates to the standardization and consistency of medical provider education interventions for our women and parents.
- Study limitations/conclusion- These were acknowledged and described accurately. The potential solutions to the findings are reasonable and may are supported by the limitations of the study within this Italian region.
Summary:
Strengths 1. The project attempts to address a critical unmet clinical and public need to improve RVV acceptance in an area with less optimal vaccine uptake
- Research design using the Health Benefits Belief Model that can provide a more rigorous model framework to address vaccine uptake
- Focusing on a specific region and highlighting that reasons for vaccine acceptance are regionally and culturally driven as opposed to a one size fits all
- Discovery that parents that are convinced of the benefits balanced with the safety concerns are at higher level of acceptance
Weaknesses
- No validation of the data collection instrument
- Small sample size of potentially culturally distinct Southern Italians
- Lack of description of the consistency of medical providers education efforts
- Largely descriptive findings with a lack of a more granular understanding of parents reasons for refusal.
Answer: We thank you for all your suggestions and we added to the discussion section a proposal to increase acceptability of RVV by all the healthcare workers involved in the pregnancy pathway.
Reviewer 4 Report
Comments and Suggestions for Authors
1) The Authors performed an interesting article entitled "Determinants of Rotavirus Vaccine Acceptance in an area with low vaccination coverage: a Case-Control Study by the Health Belief Model Questionnaire". The principal objective of the study was to examine the main reason for vaccination adherence among children’s parents in an area of Italy with low vaccination coverage. In addition, the publication has therapeutic interest, was approved by an Ethics Committee, and contains a statistical analysis of the data. However, I present some aspects that should be mentioned or improved in the manuscript.
2) What rotavirus vaccines are available in Italy? Please indicate in detail the type of vaccine, posology and administration, and pharmacodynamic properties.
3) Regarding the undesirable effects of vaccines, please indicate which are the main ones and their frequency?
4) The benefits of vaccination should be highlighted in more depth in the article.
5) There is a need to increase health literacy, in particular about the vaccination. All healthcare professionals should be involved, in particular the pharmacist as a medicinal product specialist. This aspect should be highlighted in the article.
6) The way the sample size was calculated should be explained in more detail in the article.
7) I think the title of the article should refer to the geographic area where the study was carried out.
Author Response
1) The Authors performed an interesting article entitled "Determinants of Rotavirus Vaccine Acceptance in an area with low vaccination coverage: a Case-Control Study by the Health Belief Model Questionnaire". The principal objective of the study was to examine the main reason for vaccination adherence among children’s parents in an area of Italy with low vaccination coverage. In addition, the publication has therapeutic interest, was approved by an Ethics Committee, and contains a statistical analysis of the data. However, I present some aspects that should be mentioned or improved in the manuscript.
2) What rotavirus vaccines are available in Italy? Please indicate in detail the type of vaccine, posology and administration, and pharmacodynamic properties.
Answer: Thank you for the suggestion. The information about the vaccine available in Italy has been added in the introduction section.
3) Regarding the undesirable effects of vaccines, please indicate which are the main ones and their frequency?
Answer: The main adverse effect of Rotavirus vaccine and its frequency was added in the introduction section.
4) The benefits of vaccination should be highlighted in more depth in the article.
Answer: Perceived benefits according to the HBM model adopted in the study are evaluated with two questions related to the safety and efficacy of the RVV. The discussion was modified in several parts highlighting the role of the perception of benefits that parents have on RVV and how can be used this topic in vaccine counseling.
5) There is a need to increase health literacy, in particular about the vaccination. All healthcare professionals should be involved, in particular the pharmacist as a medicinal product specialist. This aspect should be highlighted in the article.
Answer: The role of all healthcare workers including pharmacists in informing pregnant women before birth about vaccination was added in the discussion section.
6) The way the sample size was calculated should be explained in more detail in the article.
Answer: The sample size was explained in more detail as follows “The sample size was evaluated using the formula n=1,962 x proportion x (1-proportion) /accuracy. The calculation was based on the expected positive attitude of parents towards the willingness to vaccinate their children and reaching the objective of the Italian PNPV coverage, with an RVV of 85% [19,29], a confidence interval (CI) of 95%, and an accuracy of 5%. According to this formula the sample size was 196 individuals. Assuming a refusal of 10%, the final sample size was calculated as 216 individuals.”
7) I think the title of the article should refer to the geographic area where the study was carried out.
Answer: According to your suggestion the area where the study was conducted was included in the title of the study.
Round 2
Reviewer 3 Report
Comments and Suggestions for Authors
I have reviewed the authors responses and editing and thank them for this effort. The quality and relevance of the manuscript has been significantly improved. The reviewer is grateful to the group and acknowledges this important work-
Reviewer 4 Report
Comments and Suggestions for Authors
Dear Authors,
With the changes performed and the information provided, the article is now suitable for publication.